# Neoadjuvant Chemotherapy Followed by Concurrent Chemoradiation Versus Adjuvant Chemotherapy Following Concurrent Chemoradiation for Locally Advanced Cervical Cancer: A Network Meta-Analysis

**DOI:** 10.3390/cancers17020223

**Published:** 2025-01-11

**Authors:** Young Ju Suh, Dae Hyung Lee, Hee Joong Lee, Banghyun Lee

**Affiliations:** 1Department of Biomedical Sciences, Inha University College of Medicine, Incheon 22332, Republic of Korea; ysuh@inha.ac.kr; 2The Biostatistics Center, Inha University Hospital, Incheon 22332, Republic of Korea; dhlee0514@inhauh.com; 3Department of Obstetrics & Gynecology, Uijeongbu St. Mary’s Hospital, College of Medicine, The Catholic University of Korea, Seoul 11765, Republic of Korea; heejoong@catholic.ac.kr; 4Department of Obstetrics and Gynecology, Inha University Hospital, Inha University College of Medicine, Incheon 22332, Republic of Korea

**Keywords:** adjuvant chemotherapy, cervical cancer, concurrent chemoradiation therapy, neoadjuvant chemotherapy, overall survival, progression-free survival

## Abstract

The effects of neoadjuvant chemotherapy followed by concurrent chemoradiation therapy (CCRT) and adjuvant chemotherapy following CCRT on the survival of women with locally advanced cervical cancer have not been clearly compared. This study compared the effects of neoadjuvant chemotherapy followed by CCRT and adjuvant chemotherapy following CCRT on survival using a network meta-analysis to determine the optimal treatment of women with locally advanced cervical cancer. In women with locally advanced cervical cancer, neoadjuvant chemotherapy followed by CCRT had similar effects on survival to adjuvant chemotherapy following CCRT. These results provide valuable information for therapeutic strategies in women with locally advanced cervical cancer.

## 1. Introduction

The incidence and mortality of cervical cancer remain high in many low-income and middle-income countries [1]. Approximately 60% of women with cervical cancer are diagnosed with locally advanced disease [2]. Locally advanced cervical cancer (LACC) is usually defined as stages IB2 to IVA, based on the 2009 International Federation of Gynecology and Obstetrics (FIGO) staging system [3].

In women with LACC, the standard therapy is concurrent chemoradiation therapy (CCRT) with brachytherapy, consisting of external beam radiotherapy (EBRT) and concurrent platinum-containing chemotherapy [4]. Nevertheless, 25–40% of women treated by CCRT experience recurrence [5].

Many randomized controlled trials (RCTs) have investigated strategies to improve the effects of CCRT in women with LACC [6,7,8,9,10,11,12,13,14]. The effects of adjuvant chemotherapy following CCRT (CCRT + ACT) on survival compared to CCRT in RCTs performed in women with LACC are controversial [6,7,8,9,10,11]. A recent large-scale RCT showed that adjuvant chemotherapy (ACT) after CCRT in women with LACC did not improve the progression-free survival (PFS) and overall survival (OS) compared to CCRT [11]. Some meta-analyses reported inconsistent effects of CCRT + ACT on survival compared to CCRT in women with LACC [15,16,17,18]. On the other hand, some RCTs reported inconsistent effects of neoadjuvant chemotherapy followed by CCRT (NACT + CCRT) on the survival of women with LACC compared to CCRT [12,13,14]. Recently, a large-scale RCT reported that NACT + CCRT improved the PFS and OS compared to CCRT in women with LACC [13].

The optimal treatment selection in women with LACC requires a comparison of the effects of NACT + CCRT and CCRT + ACT on survival. Nevertheless, the effects of NACT + CCRT and CCRT + ACT have not been clearly compared. This study examined the effects of NACT + CCRT and CCRT + ACT on the survival of women with LACC using a network meta-analysis.

## 2. Materials and Methods

Ethical approval was not required because anonymous aggregate data were used.

### 2.1. Search Strategy

The PubMed, Medline, and Embase databases in October 2024 were searched to include pertinent studies using combinations of the following keywords: cervical cancer AND (chemoradiotherapy OR chemoradiation OR radiochemotherapy) AND (NACT OR ACT) AND randomized trial (Appendix A). Additional relevant studies not identified by these database searches were found by examining the references from the selected clinical studies and review articles.

### 2.2. Selection Criteria

The study selection was based on the PRISMA 2020 statement [19]. The inclusion criteria were studies of the following: histologically diagnosed cervical cancer, newly diagnosed LACC, use of CCRT, use of NACT or ACT, and RCT. The exclusion criteria were non-RCT, retrospective studies, review articles, editorials, letters, abstracts, protocols, case reports, or irrelevant studies. Only studies with the most adequate data were selected to avoid, including duplicate information when studies included overlapping groups of patients.

### 2.3. Data Extraction and Outcomes of Interest

Two investigators independently extracted the data of interest from studies using the checklist.

Discrepancies between investigators were resolved by discussion. The eligible population was classified into three groups (NACT + CCRT, CCRT + ACT, and CCRT [the control group]) based on whether they had received NACT, ACT, or CCRT alone. The data retrieved from these studies were the names of the first author and the study, year of publication, study design, stage, median follow-up time, number of participants, treatment methods, number of disease progressions or deaths, and statistical data of PFS and OS. PFS was defined as the time from randomization to disease progression or death from any cause. OS was defined as the time from randomization to death from any cause.

### 2.4. Statistical Analyses

A network meta-analysis uses a multivariate random-effect model with a frequentist framework [20]. The hazard ratios (HRs) were considered summary estimates of the effect sizes of the treatment responses for cervical cancer progression and mortality. The I^2^ statistic and Cochran’s Q statistic, which are heterogeneity indices, were used to estimate whether a dispersion occurred among the HRs across the included studies.

The rank probabilities of the treatment efficacy were estimated using the surface under the cumulative ranking curve (SUCRA) method [21]. The treatment ranks were evaluated based on the HRs derived from pairwise comparisons within the network meta-analysis. Since HRs are relative measures, each treatment’s rank was determined by aggregating its performance across all pairwise comparisons within the network. The ranking process involved 1000 simulations, during which the relative performance of each treatment was evaluated to calculate the probabilities of being ranked as “Best”, “2nd”, or “Worst”. These rankings reflect the relative efficacy of treatments across the entire network rather than the absolute effect sizes. The SUCRA score was calculated as the cumulative probability of each treatment group being ranked as “Best”, “2nd”, or “Worst”. The SUCRA values ranged from 0 to 100%, with a larger area under the SUCRA curve indicating a higher ranking.

R software (Version 4.1.1, ‘netmeta’ package; R Foundation for Statistical Computing, Vienna, Austria) and STATA software Version 18 (StataCorp LLC, College Station, TX, USA) were used for the statistical analysis.

## 3. Results

### 3.1. Included Studies: Search Results, Characteristics, and Assessments of Risk Bias

The literature search initially identified 368 potentially relevant studies, and six RCTs that met the selection criteria were ultimately identified (Figure 1).

Table 1 lists the characteristics of the included studies, and Appendix A presents the results of the assessments of risk bias.

In the included studies, NACT was followed by CCRT [12,13,14], and ACT was performed after CCRT [8,10,11]. The included studies enrolled 2446 women with LACC (378 from three studies on NACT + CCRT, 852 from three studies on CCRT + ACT, and 1216 controls treated with CCRT) (Table 1). The eligible population diagnosed with LACC included FIGO 2009 stages ⅠB1 with nodal involvement to stage IVA or stages IIB to IVA [8,10,11,12,13,14].

### 3.2. Network Meta-Analyses for PFS and OS

Figure 2 shows network plots of the pooled included studies on PFS and OS in the eligible populations.

Three treatment arms of NACT + CCRT, CCRT + ACT, and CCRT were identified in the plots. Significant evidence in heterogeneity was observed between the studies for the comparison between NACT + CCRT and CCRT (I^2^ = 79% and *p* = 0.009 in PFS; I^2^ = 83.1% and *p* = 0.003 in OS) or between CCRT + ACT and CCRT (I^2^ = 82.6% and *p* = 0.003 in PFS; I^2^ = 53.6% and *p* = 0.116 in OS).

Figure 3 presents the network meta-analysis results for the PFS and OS. 

For PFS, NACT + CCRT exhibited a similar hazard for cervical cancer progression compared to the CCRT. In addition, the hazard of cervical cancer progression for CCRT + ACT exhibited similar hazards compared to the CCRT. An indirect comparison revealed no significant difference between NACT + CCRT and CCRT + ACT. For OS, NACT + CCRT exhibited a similar hazard for death compared to the CCRT. In addition, the hazard of death for CCRT + ACT was similar to that when the CCRT was used. An indirect comparison estimated no significant difference between NACT + CCRT and CCRT + ACT. Figure 4 presents the Forest plots.

Table 2 lists the SUCRA values for the treatments, and Figure 5 shows the SUCRA curve for the treatments.

For PFS, the SUCRA analysis (presented in Figure 5) showed that the surface area for the CCRT + ACT treatment was the largest among all the treatments, indicating that it had the highest likelihood of being the most effective treatment. The larger surface area for CCRT + ACT reflects its superior ranking compared to the other treatments for PFS. The probability of CCRT + ACT treatment being the best was 57.2%, and the cumulative probability of it being at least the second best was 86.8% (calculated as 57.2% + 29.6%). For OS, the probability of the CCRT + ACT treatment being the best was approximately 50.9%, and the cumulative probability of it being at least the second best was 77.7% (calculated as 50.9% + 26.8%). In the SUCRA analysis (shown in Figure 5), the surface area for the CCRT + ACT treatment was the largest among all the treatments, indicating that it had the highest likelihood of being the most effective treatment for OS.

For the PFS and OS, CCRT + ACT had the highest SUCRA value, suggesting that this method may be a better treatment option for preventing cervical cancer progression and death. For PFS, the SUCRA values increased in the order of CCRT + ACT, CCRT, and NACT + CCRT. For OS, the SUCRA values were similar in NACT + CCRT and CCRT.

## 4. Discussion

No study directly compared the effects of NACT + CCRT and CCRT + ACT on the survival of women with LACC. In studies with different designs, head-to-head comparisons are limited to determine the effects of therapeutic strategies. A network meta-analysis provides an indirect comparison in such situations. This paper reports the results of a study performed using this technique that indirectly compared the effects of NACT + CCRT and CCRT + ACT in women with LACC. NACT + CCRT had indistinguishable effects on the PFS and OS compared to CCRT + ACT. Moreover, a direct comparison between NACT + CCRT and CCRT and between CCRT + ACT and CCRT showed similar effects on survival. On the other hand, the SUCRA values showed that CCRT + ACT had the highest probability of being the most effective treatment in terms of PFS and OS in women with LACC.

Regarding the treatment strategies for cervical cancer, surgery is commonly used in early stage cancers, and radiotherapy is a major treatment of cervical cancer. Chemotherapy is used as an adjuvant therapy after surgery to reduce the risk of recurrence, in combination with radiotherapy to enhance the effects of radiotherapy, and as a single treatment in locally advanced disease [23]. Many RCTs have investigated the effects of CCRT, radiotherapy, or surgery with neoadjuvant or adjuvant therapies in women with LACC [6,7,8,9,10,11,12,13,14,24,25,26,27,28,29]. A recent network meta-analysis of RCTs showed that CCRT and CCRT + ACT are likely to be the optimal treatments in terms of the PFS and OS for women with LACC by comparing the nine different therapeutic strategies: CCRT alone, CCRT + ACT, CCRT followed by surgery, NACT + CCRT, radiotherapy alone, ACT following radiotherapy, radiotherapy followed by surgery, surgery alone, and NACT followed by surgery [29].

Many RCTs compared the effects of CCRT + ACT or NACT + CCRT with CCRT in women with LACC [6,7,8,9,10,11,12,13,14]. Previous RCTs and meta-analyses performed in women with LACC showed that CCRT + ACT had inconsistent effects on survival compared to CCRT [6,7,8,9,10,11]. In particular, the recent largest-scale RCT (OUTBACK study, *n* = 919) showed that the PFS and OS in women with LACC were similar in CCRT + ACT and CCRT [11]. In this study, including three RCTs [8,10,11], CCRT + ACT showed a similar PFS and OS to the CCRT in women with LACC. On the other hand, in three RCTs performed on women with LACC, NACT + CCRT had inconsistent effects on survival compared to CCRT [12,13,14]. One RCT (*n* = 107) reported a higher PFS and OS in CCRT than NACT + CCRT [12]. A recent RCT (*n* = 146) reported similar PFS and higher OS in NACT + CCRT compared to CCRT [14]. The recent large-scale RCT (the INTERLACE study, *n* = 500) reported that NACT + CCRT had a higher PFS and OS than CCRT (abstract only) [13]. In the present study, including those studies, NACT + CCRT showed a similar PFS and OS to the CCRT in women with LACC.

Recently, two large-scale RCTs that evaluated the effects of NACT + CCRT or CCRT + ACT compared to CCRT were reported [11,13]. In women with LACC, however, the effects of NACT + CCRT and CCRT + ACT on survival compared to the CCRT are still inconclusive [30,31]. At this time, selecting the optimal treatment in women with LACC requires comparing the effects of NACT + CCRT and CCRT + ACT on survival. A recent network meta-analysis of RCTs performed on women with LACC [29] reported lower PFS and higher OS in CCRT + ACT compared to NACT + CCRT (OR 3.69 and 95% CI 1.18–11.58 for PFS; OR 0.28 and 95% CI 0.08–0.98 for OS), higher PFS and similar OS in CCRT compared to CCRT + ACT (Odds ratio (OR) 0.60 and 95% CI 0.38–0.96 for PFS; OR 0.84 and 95% CI 0.53–1.31 for OS), and similar PFS and higher OS in CCRT compared to NACT + CCRT (OR 2.21 and 95% CI 0.78–6.28 for PFS; OR 0.23 and 95% CI 0.07–0.75 for OS). Among those treatments, CCRT and CCRT + ACT had higher SUCRA values for the PFS and OS than NACT + CCRT (CCRT vs. CCRT + ACT vs. NACT + CCRT: 80.8% vs. 98.7% vs. 46.7% in PFS and 90% vs. 73.5% vs. 6.3% in OS). Nevertheless, that study compared various therapeutic strategies used in women with LACC and included only four studies to examine CCRT + ACT and one study for NACT + CCRT. Therefore, a network meta-analysis was performed using the currently available RCTs that reported the effects of NACT + CCRT or CCRT + ACT in women with LACC. This study showed that in women with LACC, the PFS and OS were similar in NACT + CCRT and CCRT + ACT: between NACT + CCRT and CCRT and between CCRT + ACT and CCRT. On the other hand, CCRT + ACT had higher SUCRA values, and NACT + CCRT and CCRT had lower SUCRA values, suggesting that CCRT + ACT might be the most effective treatment for the PFS and OS among NACT + CCRT, CCRT, and CCRT + ACT. In the present study, discrepancies between the results of the indirect and direct comparisons and SUCRA values require further clarification.

The present study used a network meta-analysis to compare the effects of NACT + CCRT and CCRT + ACT on the survival of women with LACC, but this study had the following limitations. First, only three RCTs compared the effects of NACT + CCRT on survival with CCRT with LACC. In those studies, two studies were not large-scale studies, and one large-scale study (the INTERLACE) was reported as an abstract. Nevertheless, this study included all the studies currently available. Second, this network meta-analysis compared only three RCTs of NACT + CCRT and only three RCTs of CCRT + ACT. Therefore, the small number of studies included may limit the significance of this study. Third, high heterogeneities were observed between the RCTs for the comparison between NACT + CCRT and CCRT and between RCTs for the comparison between CCRT + ACT and CCRT. Therefore, this study used the random-effect model for the network meta-analysis.

## 5. Conclusions

Although this study was limited by the comparisons between RCTs with different designs, the indirect comparisons made using a network meta-analysis approach suggest that the effects of NACT + CCRT and CCRT + ACT were not different on the PFS and OS in women with LACC. Moreover, this study showed that CCRT with and without chemotherapy (neoadjuvant or adjuvant) had similar effects on the survival of women with LACC. On the other hand, this study demonstrated that based on the SUCRA value, CCRT + ACT might be a better strategy in terms of survival in women with LACC. These results provide valuable information for therapeutic strategies in women with LACC. Future studies will be needed to obtain crucial insights into the effects of NACT + CCRT, CCRT, and CCRT + ACT in women with LACC.

## Figures and Tables

**Figure 1 cancers-17-00223-f001:**
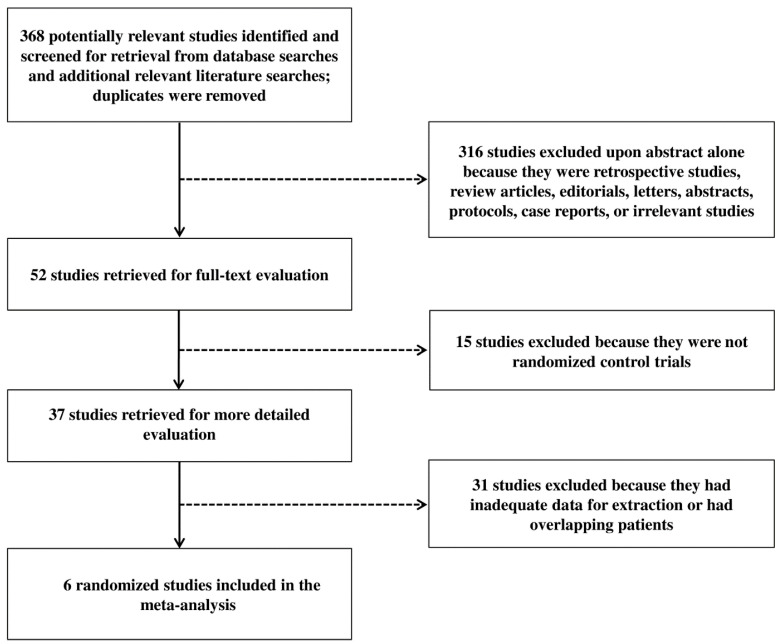
Flow chart of study selection.

**Figure 2 cancers-17-00223-f002:**
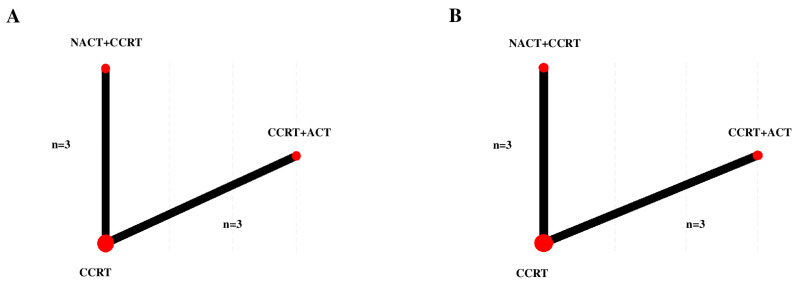
Network plots of treatments for the PFS and OS. (**A**) PFS; (**B**) OS. The size of the three nodes (treatments) increased as the number of studies included in the corresponding nodes increased, and the lines connecting two nodes were thickened as the number of studies comparing the two treatments increased [22]. CCRT, concurrent chemoradiation therapy; CCRT + ACT, adjuvant chemotherapy following concurrent chemoradiation therapy; NACT + CCRT, neoadjuvant chemotherapy followed by concurrent chemoradiation therapy; OS, overall survival; and PFS, progression-free survival.

**Figure 3 cancers-17-00223-f003:**
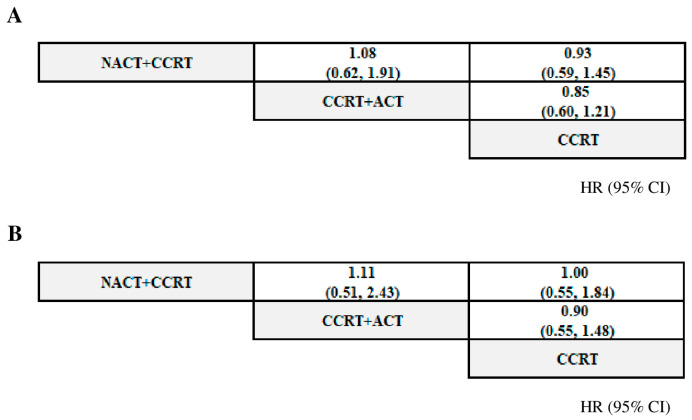
League tables of the treatments for the PFS and OS. (**A**) PFS; (**B**) OS. The hazard ratio (HR) of the upper left treatment (intervention) vs. lower right (comparator) was estimated. CCRT, concurrent chemoradiation therapy; CCRT + ACT, adjuvant chemotherapy following concurrent chemoradiation therapy; CI, confidence interval; NACT + CCRT, neoadjuvant chemotherapy followed by concurrent chemoradiation therapy; OS, overall survival; and PFS, progression-free survival.

**Figure 4 cancers-17-00223-f004:**
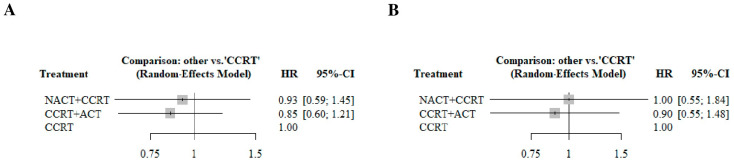
Forest plots of the treatments for the PFS and OS. (**A**) PFS, (**B**) OS. CCRT, concurrent chemoradiation therapy; CCRT + ACT, adjuvant chemotherapy following concurrent chemoradiation therapy; CI, confidence interval; NACT + CCRT, neoadjuvant chemotherapy followed by concurrent chemoradiation therapy; HR, hazard ratio; OS, overall survival; PFS, progression-free survival.

**Figure 5 cancers-17-00223-f005:**
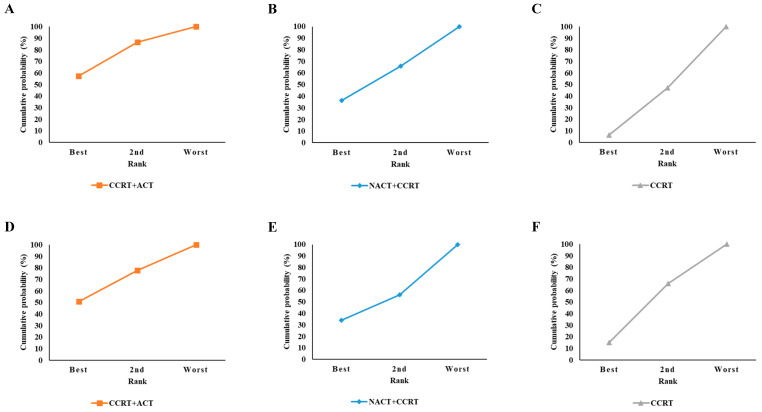
SUCRA curves of the treatments for the PFS and OS. The treatments for the PFS: (**A**) CCRT + ACT, (**B**) NACT + CCRT, and (**C**) CCRT; the treatments for the OS: (**D**) CCRT + ACT, (**E**) NACT + CCRT, and (**F**) CCRT. CCRT, concurrent chemoradiation therapy; CCRT + ACT, adjuvant chemotherapy following concurrent chemoradiation therapy; NACT + CCRT, neoadjuvant chemotherapy followed by concurrent chemoradiation therapy; OS, overall survival; PFS, progression-free survival; and SUCRA, the estimated surface under the cumulative ranking probabilities.

**Table 1 cancers-17-00223-t001:** Characteristics of the included studies in which women with LACC underwent NACT + CCRT or CCRT + ACT.

Authors	Design	Population	Number of Participants	Treatment Arms	PFS				OS			
					Event (n)	HR	95% CI	*p* Value	Event (n)	HR	95% CI	*p* Value
da Costa SCS et al. (2019) [12] CIRCE	RCT, Phase 2	Stage IIB—IVA, median follow-up time: >48.7 months	NACT + CCRT: 55 CCRT: 52	NACT + CCRT: cisplatin 50 mg/m^2^ IV on day 1 plus gemcitabine 1000 mg/m^2^ IV on day 1 and day 8 every 3 weeks for 3 cycles, and then cisplatin 40 mg/m^2^ weekly for 6 weeks with concurrent EBRT 45–50.4 Gy, followed by brachytherapy. CCRT: cisplatin and concurrent EBRT followed by brachytherapy (dosing same as for neoadjuvant chemotherapy + CCRT).	NACT + CCRT: 33 CCRT: 21	1.84	1.04–3.26	0.03	NACT + CCRT: 22 CCRT: 7	2.79	1.29–6.01	0.01
McCormack M et al. (2023) [13], INTERLACE	RCT, Phase 3 (Abstract)	Stage IB1 lymph node-positive—IVA (IB1/2: 9%; II: 77%), median follow-up time: 64 months	NACT + CCRT: 250 CCRT: 250	NACT + CCRT: weekly carboplatin AUC 2 IV plus weekly paclitaxel 80 mg/m^2^ IV for 6 weeks, and then cisplatin 40 mg/m^2^ weekly for 5 weeks with concurrent EBRT 40–50.4 Gy, followed by brachytherapy. CCRT: cisplatin and concurrent EBRT followed by brachytherapy (dosing same as for neoadjuvant chemotherapy + CCRT).	NACT + CCRT: 183 CCRT: 160	0.65	0.46–0.91	0.01	NACT + CCRT: 200 CCRT: 180	0.61	0.40–0.91	0.04
Li F et al. (2024) [14]	RCT, Phase 2	Stage IIB—IVA, median follow-up time: 21 months	NACT + CCRT: 73 CCRT: 73	NACT + CCRT: cisplatin 60–80 mg/m^2^ IV plus paclitaxel 135–175 mg/m^2^ IV every 3 weeks for 2 cycles, and then cisplatin 40 mg/m^2^ weekly for 6 weeks with concurrent EBRT 50.4 Gy, followed by brachytherapy. CCRT: cisplatin and concurrent EBRT followed by brachytherapy (dosing same as for neoadjuvant chemotherapy + CCRT).	NACT + CCRT: 14 CCRT: 15	0.72	0.31–1.68	0.34	NACT + CCRT: 8 CCRT: 14	0.69	0.11–0.92	0.02
Dueñas-González A et al. (2011) [8]	RCT, Phase 3	Stage IIB—IVA, median follow-up time: 46.9 months	CCRT + ACT: 259 CCRT: 256	CCRT + ACT: cisplatin 40 mg/m^2^ IV and gemcitabine 125 mg/m^2^ IV weekly for 6 weeks with concurrent EBRT 50.4 Gy, followed by brachytherapy, and then cisplatin 50 mg/m^2^ IV on day 1 plus gemcitabine 1000 mg/m^2^ IV on days 1 and 8 every 3 weeks for 2 cycles. CCRT: cisplatin and concurrent EBRT followed by brachytherapy (dosing is the same as CCRT + adjuvant chemotherapy).	CCRT + ACT: 75 CCRT: 97	0.68	0.49–0.95	0.02	CCRT + ACT: 65 CCRT: 92	0.68	0.49–0.95	0.02
Tovanabutra C et al. (2021) [10], ACTLACC	RCT, Phase 2	Stage IIB—IVA without para-aortic lymph node enlargement, median follow-up time: 40.9 months	CCRT + ACT: 130 CCRT: 129	CCRT + ACT: cisplatin 40 mg/m^2^ IV weekly for 6 weeks with concurrent EBRT 45–50.4 Gy, followed by brachytherapy, and then carboplatin AUC 5 IV plus paclitaxel 175 mg/m^2^ IV every 4 weeks for 3 cycles. CCRT: cisplatin and concurrent EBRT followed by brachytherapy (dosing is the same as CCRT + adjuvant chemotherapy).	CCRT + ACT: 42 CCRT: 39	1.22	0.80–1.87	0.35	CCRT + ACT: 39 CCRT: 30	1.27	0.76–2.10	0.34
Mileshkin LR et al. (2023) [11], OUTBACK	RCT, Phase 3	Stage 1B1 lymph node-positive—IVA (1B1, 1B2 or IIA: 33%), median follow-up time: 60 months	CCRT + ACT: 463 CCRT: 456	CCRT + ACT: cisplatin 40 mg/m^2^ IV weekly for 5 weeks with concurrent EBRT 45–50.4 Gy, followed by brachytherapy, and then carboplatin AUC 5 IV plus paclitaxel 155 mg/m^2^ IV every 3 weeks for 4 cycles. CCRT: cisplatin and concurrent EBRT followed by brachytherapy (dosing is the same as CCRT + adjuvant chemotherapy).	CCRT + ACT: 171 CCRT: 173	0.86	0.69–1.08	0.58	CCRT + ACT: 130 CCRT: 132	0.90	0.70–1.17	0.81

CCRT, concurrent chemoradiation therapy; CCRT + ACT, adjuvant chemotherapy following concurrent chemoradiation therapy; CI, confidence interval; EBRT, external beam radiation therapy; HR, hazard ratio; LACC, locally advanced cervical cancer; NACT + CCRT, neoadjuvant chemotherapy followed by concurrent chemoradiation therapy: OS, overall survival; PFS, progression-free survival; and RCT, randomized controlled trial.

**Table 2 cancers-17-00223-t002:** SUCRA values of the treatments for the PFS and OS.

		Treatment Efficacy				
	Treatment	Best	2nd	Worst	SUCRA	Rank
PFS						
	CCRT + ACT	57.2%	29.6%	13.2%	72.0%	1
	CCRT	6.4%	40.8%	52.8%	51.2%	2
	NACT + CCRT	36.4%	29.6%	34.0%	26.8%	3
OS						
	CCRT + ACT	50.9%	26.8%	22.3%	64.3%	1
	NACT + CCRT	33.9%	22.3%	43.8%	45.1%	2
	CCRT	15.2%	50.9%	33.9%	40.7%	3

CCRT, concurrent chemoradiation therapy; CCRT + ACT, adjuvant chemotherapy following concurrent chemoradiation therapy; NACT + CCRT, neoadjuvant chemotherapy followed by concurrent chemoradiation therapy; OS, overall survival; PFS, progression-free survival; and SUCRA, the estimated surface under the cumulative ranking probabilities.

## Data Availability

All the data generated or analyzed during this study are included in this published article and its Appendix A files.

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
