# Peer review of "Neoadjuvant Chemotherapy Followed by Concurrent Chemoradiation Versus Adjuvant Chemotherapy Following Concurrent Chemoradiation for Locally Advanced Cervical Cancer: A Network Meta-Analysis"

_cancers, 2025, doi:10.3390/cancers17020223_

Round 1

Reviewer 1 Report

Comments and Suggestions for Authors

Dear Authors,

Thank you for the opportunity to read this interesting manuscript.                                 Systematic reviews of well-designed and well-conducted randomized controlled trials  provide the best evidence to support treatment decisions. The development of a new statistical method, network meta-analysis, enables the simultaneous comparison of multiple interventions. The Authors compared the effects of NACT+CCRT and CCRT+ACT on survival using network meta-analysis to select the optimal treatment in women with LACC based on indirect comparison. In the conclusion they state that in women with LACC, NACT+CCRT had no different effects on the PFS and OS compared to CCRT+ACT, even though CCRT+ACT had a higher SUCRA value.

Below are some comments, I hope they are useful:

Introduction:

Lines 50-51: …”In women with LACC, the standard therapy is concurrent chemoradiation therapy (CCRT) with vaginal brachytherapy”  my commentary: Brachytherapy in patient with intact uterine in simple terms, means inserting applicators into the uterus and vagina; so the use of the term "vaginal brachytherapy" is incorrect.

Figure 2: the figure caption is inaccurate – part A probably refers to PFS, part B to OS.

Results:

Below Figure 3, which presents the results of the network meta-analysis in tabular form, I would add the information that the results should be read in order from left to right.

Table 2: The authors compared the treatment methods by ranking them, i.e. determining the probability of each treatment method occupying a specific place in the network. In reporting the rankogram, they used the surface under the cumulative ranking curve (SUCRA). Given that not every potential reader may have knowledge of network analysis, in my opinion it would be helpful to include a figure with a graphical way of presenting the results instead of a table or next to a table .

Reviewer 2 Report

Comments and Suggestions for Authors

Well designed and well-presented meta-analysis of RCT regarding the best treatment option for women with locally advanced cervical cancer.

The analysis showed that there were no significant differences between NACT followed by CCRT and CCRT followed by ACT.

1-The main question addressed by the research is:  comparison between NACC followed by CCRT and CCRT followed by ACT in locally advanced cervical carcinoma.    2- The topic is a meta-analysis of RCT, it is original and relevant to the management of LACC.   3- Meta-analysis of RCT is the best evidence so far on the topic.   4- The authors followed the methodology known for RCT. This is not a new study, but an analysis of RCT.   5- The conclusion from this study is: NACC followed by CCRT is not significantly different from CCRT followed by ACT.   6- The references are relevant to the topic.   7- As I am not an expert in statistics, I would recommend statistical revision of the results and tables sections.

Reviewer 3 Report

Comments and Suggestions for Authors

This study compared the effects of neoadjuvant chemotherapy followed by CCRT and adjuvant chemotherapy following CCRT on survival using network meta-analysis to determine the optimal treatment of women with locally advanced cervical cancer, and concluded that CCRT+ACT may be a better treatment option for preventing cervical cancer progression and death.

In the international Phase 3 OUTBACK trial, adjuvant chemotherapy with carboplatin and paclitaxel after chemoradiotherapy did not improve progression-free survival or overall survival of cervical cancer, which is contradictory with the conclusion of this study.
